# Association of Polymorphism within the Putative miRNA Target Site in the 3′UTR Region of the *DRD2* Gene with Neuroticism in Patients with Substance Use Disorder

**DOI:** 10.3390/ijerph19169955

**Published:** 2022-08-12

**Authors:** Agnieszka Boroń, Małgorzata Śmiarowska, Anna Grzywacz, Krzysztof Chmielowiec, Jolanta Chmielowiec, Jolanta Masiak, Tomasz Pawłowski, Dariusz Larysz, Andrzej Ciechanowicz

**Affiliations:** 1Department of Clinical and Molecular Biochemistry, Pomeranian Medical University in Szczecin, Aleja Powstańców Wielkopolskich 72 St., 70-111 Szczecin, Poland; 2Department of Pharmacokinetics and Therapeutic Drug Monitoring, Pomeranian Medical University in Szczecin, Aleja Powstańcόw Wielkopolskich 72 St., 70-111 Szczecin, Poland; 3Independent Laboratory of Health Promotion, Pomeranian Medical University in Szczecin, Aleja Powstańcόw Wielkopolskich 72 St., 70-111 Szczecin, Poland; 4Department of Hygiene and Epidemiology, Collegium Medicum, University of Zielona Góra, Zyty 28 St., 65-046 Zielona Gora, Poland; 5Second Department of Psychiatry and Psychiatric Rehabilitation, Medical University of Lublin, Głuska 1 St., 20-059 Lublin, Poland; 6Division of Psychotherapy and Psychosomatic Medicine, Wroclaw Medical University, Wyb. L. Pasteura 10 St., 50-367 Wroclaw, Poland; 7109 Military Hospital with Cutpatient Cinic in Szczecin, Piotra Skargi 9-11 St., 70-965 Szczecin, Poland

**Keywords:** addiction, dual diagnosis, *DRD2*, gene, neuroticism, personality, rs6276, SUD

## Abstract

The study aims at looking into associations between the polymorphism rs6276 that occurs in the putative miRNA target site in the 3′UTR region of the *DRD2* gene in patients with substance use disorder (SUD) comorbid with a maniacal syndrome (SUD MANIA). In our study, we did not state any essential difference in *DRD2* rs6276 genotype frequencies in the studied samples of SUD MANIA, SUD, and control subjects. A significant result was found for the SUD MANIA group vs. SUD vs. controls on the Neuroticism Scale of NEO FFI test, and *DRD2* rs6276 (*p* = 0.0320) accounted for 1.7% of the variance. The G/G homozygous variants were linked with lower results on the neuroticism scale in the SUD MANIA group because G/G alleles may serve a protective role in the expression of neuroticism in patients with SUD MANIA. So far, there have been no data in the literature on the relationship between the miRSNP rs6276 region in the *DRD2* gene and neuroticism (personal traits) in patients with a diagnosis of substance use disorder comorbid with the affective, maniacal type disturbances related to SUD. This is the first report on this topic.

## 1. Introduction

Substance use disorder (SUD), which refers to refers to substance abuse and dependence [1,2], belongs to the category of neuropsychiatric disorders and may be described as a recurring desire to revert to an addictive substance regardless of its harmful consequences [3]. Therefore, SUD is considered one of the most common mental disturbances. Among the risk factors for SUD susceptibility are male sex, a history of addiction in the family, younger age, and the coexistence of psychiatric illnesses such as affective diseases or post-traumatic stress disorder [4]. The etiology of SUD is complex and multifactorial [5] and remains under consideration because numerous genes and environmental factors are involved in this process, as seen in other neuropsychiatric disorders [6]. Additionally, the mode of inheritance is composed of the incomplete penetrance and phenocopies, variable genetic heterogeneity, and expressivity or polygenicity, as well as the phenomenon of epistasis [7]. The heritability of SUD, which may be a result of the genetic factors, is estimated to stand at 40–60% of all the observed variations. The strongest relationship is to cocaine (72%) and the lowest to hallucinogens (39%) [8]. The heritability for alcohol has been estimated to stand at 50% and opioids at 23–54% for all SUD groups [9]. Patients are in great need of a solution to SUD pathogenesis in order to be given better prevention and new patterns of treatment [10]. A more profound study and a deeper knowledge of the genes engaged in SUD determination would be very helpful.

Patients with SUD experience mania. Too few double diagnoses exist; therefore, this problem is discussed in our research.

Mania is described as a mental disorder characterized by changes in mood elevation or dysphoria with increased energy and activity, over-sexuality, racing thoughts, and a decreased need for sleep and eating, and it is often accompanied by delusions of grandiosity and omnipotence [11]. Kraepelin was the first to frame the concept of unipolar mania [12]. Although, in the latest diagnostic criteria based on DSM-5 (Diagnostic and Statistical Manual of Mental Disorders of the American Psychiatric Association 5th Revision), unipolar mania is determined as a subtype of the bipolar I disorder, according to ICD-11 (International Classification of Diseases 11th Revision), it is classified as the bipolar type I disorder, unspecified. To date, little research into unipolar mania has been performed. The available studies are based on differing criteria due to the nonexistence of a consensus for the most proper definition of unipolar mania. Although most of the studies relate to the absence of a depressive episode, there is a group of patients who experience so-called “mixed” episodes. These patients have a proven history of affective disorder, but they experience the criteria of the minimum number of manic episodes [13]. The available data on unipolar mania have shown that patients diagnosed with unipolar mania are more prone/sensitive to falling into illness at a younger age and show more psychotic symptoms, but, on the other hand, they manifest fewer symptoms of suicidality and comorbid anxiety [14,15]. Angst et al. [14] found that in adults with unipolar mania, the percentage of drug users is lower than in those with type I bipolar disorder (BP-I). This observation is in opposition to the findings of Grobler et al. [16], who believed that cannabis abuse was more frequent in patients with unipolar mania than in those with both mania and depressive episodes. 

In contrast, data collected from smaller clinical observations and based on case reports imply an association between the development of maniacal psychosis and exposure to stimulants [17,18,19,20,21,22,23,24]. Psychostimulants in their typical action increase the dopaminergic activity and thereby cause symptoms that are found not only in mania but also in healthy individuals [25]. On the other hand, dopaminergic gene variants are involved in, among other things, the ethicology of schizophrenia and affective and substance abuse disorders [25,26,27,28,29,30,31,32].

The clinical history of addiction does not seem to show enough evidence to differentiate unipolar mania episodes from the disorders of bipolar to psychoactive (SPAs), as SPA might play a role in all of them as a tool for regulating mood disturbances. Consequently, maniacal episodes seem to not be the only exogenous syndromes due to SPA intoxication, independently of real affective disease. SPA may be a trigger mechanism that indicates the real endogenous uni- or bipolar disorder. Therefore, the diagnostic process is rather complicated and unclear, especially when it comes to double diagnosis. Some data have proved that patients diagnosed with bipolar disorder have a higher exposure to substance abuse in interviews [33,34], but in another study, no difference was reported [35]. Higher amphetamine and cannabis use has been observed in people with maniac disorder of the unipolar type [36]. These differentiated findings may be derived from the lack of coherent criteria of the classification in studies in patients with unipolar mania, especially in these with additional psychotic symptoms who were more predisposed to substance addiction [14,16,36].

Open-field tests based on animal models of mania rely on the effects exerted by the administration of stimulants on the locomotion behavior in rodents [37,38,39,40]. These tests showed that the administration of stimulants (d-amphetamine) resulted in biochemical changes of dopamine (DA) neurotransmission with its increased release, reuptake inhibition, or decreased degradation in a synaptic space by the monoamine oxidase (MAO) enzyme [41,42], which, ultimately, produced an increased psychomotor activity including agitation [43,44,45]. Moreover, cognitive functioning has also been found to be significantly impaired in animal models of mania induced by psychostimulants [46,47,48]. Several preclinical experiments with mood stabilizers and potential new targets for manic-like behaviors, such as d-amphetamine, lisdexamfetamine dimesylate (LDX), GBR12909, and fenproporex, have been carried out up to now. Their reversal and observations of prevention of mania are well documented [49,50,51,52,53].

These pharmacological agents commonly enhance dopamine activity, which may also be of great significance in the pathophysiology of mania in human individuals [40,54].

Additionally, in animal models of mania, the acute administration of quinpirole, which is a D2R (dopamine receptor D2) agonist, at certain doses, induced hyperactivity [55], while PET studies confirmed that the numbers of D2Rs in mania may be elevated [56].

D2Rs belong to the family of G protein- linkage receptors and are mainly located on postsynaptic dopaminergic neurons of the meso–corticolimbic pathway. They are probably involved in reward-mediating regulations [57]. The *DRD2* gene (gen for dopamine D2 receptor) encodes two isoforms that are molecularly and functionally different [58]. The affinity for dopamine of the D2Rs is 10- to 100-fold greater than that described in the D1Rs (dopamine D1 receptors) family. Therefore, the balance between D2R and D1R receptor signaling may alter the availability of extracellular dopamine concentrations. D2R occurs in two isoforms, S and L, which differ in an effect of a 29 AA insertion in the third intracellular loop on L form [59]. The isoform S of D2R dominates in the body cells and distant projections of the dopaminergic axons in mesencephalon, whereas the L isoform is mainly located on dopaminergic neuron terminals in the striatum and nucleus accumbens [60,61,62,63].

Dopamine signaling through D2Rs is responsible for physiological functions related to the locomotion function, which is involved in nigrostriatal hormone production with the tubo-infundibular pathway and drug abuse with the meso–cortico–limbic pathway [64]. To date, the role of drugs with affinity to D2Rs in their psychotic, parkinsonian, cognitive, motivation, and behavioral processes has been well, but not sufficiently, documented [65,66].

It has been observed that persons who regularly take cocaine, methamphetamine, and opiates in a chronic and addictive pattern, as well as alcoholics, may develop long-term decreases in the expression of D2Rs, which leaves them with a reduced sensitivity to nonhabitual rewards [67].

The *DRD2* gene, which is located on 11 q22–q24, is one of the most studied genes involved in different behavioral disorders such as ADHD (Attention Deficit and Hyperactivity Disorder), as well as in endogenous diseases such as schizophrenia and many neurodegenerative movement disorders [68,69].

Some somatic changes connected with an elevated blood pressure or even hypertension have been explained by the presence of common single-nucleotide polymorphisms, such as SNPs rs6276, rs6277, and rs1800497 in the *DRD2* gene, which cause decreased D2R expression. These effects relate to decreased *DRD2* mRNA stability, as well as the synthesis of the receptor [70,71,72,73].

Little is known about the associations of genetic polymorphisms and a group of the psychiatric disturbances. A systematic review was conducted to search for some possible candidates. The data from Marco et al. [74] based on selected studies, referred to patients with bipolar disorder comorbid with substance abuse and showed 66 polymorphisms in 29 genes that had been previously investigated [74]. Among those with potential genetic value in coexisting of bipolar disorder and SUD are rs11600996 (*ARNTL*), rs228642/rs228682/rs2640909 (*PER3*), PONQ192R (*PON1*), rs945032 (*BDKRB2*), rs1131339 (*NR4A3*), and rs6971 (*TSPO*) [74].

The rs6276 SNP in the 3′ untranslated region was described as connected with various alcohol phenotypes, perhaps as a consequence of a lower transcription or translation rate, which finally results in a decreased D2R availability [69,75].

Moreover, a large number of different psychological measures have been established as risk factors for the occurrence of mania episodes, independent of etiology. Among them there are the General Behavior Inventory (GBI) [76], the Temperament Evaluation of Memphis, Pisa, Paris, and San Diego [77], the Hypomanic Personality Scale (HPS), the Hypomania Check List (HCL-32) [78,79], and the Mood Disorder Questionnaire [80]. Each of these is used to assess tendencies for subsyndromal manic symptoms and specific personality traits believed to depend on bipolar disorder (BD). These gathered findings let investigators assume that a risk of mania might be linked to the early stages of the disorder, as well as the genetic polymorphic originality, which commonly participates in dopamine regulation [80,81,82,83].

Our study aimed to investigate associations between the polymorphism rs6276’s putative site in the miRNA of the 3′UTR region of the *DRD2* gene in patients with substance use disorder comorbid with exacerbated clinical mania and selected personality features.

## 2. Materials and Methods

### 2.1. Materials

The volunteers participating in the study comprised 585 males. They were divided into three groups: patients with a coexisting diagnosis of substance use disorder (SUD) and maniac episodes (SUD MANIA; *n* = 89; mean age = 27.9, SD = 6.6) or purely polysubstance use disorder without mania (SUD; *n* = 195; mean age = 28.3, SD = 6.4) and healthy controls (*n* = 301; mean age = 22.1, SD = 4.6) (Figure 1).

Concerning the SUD MANIA group, only male patients were recruited in order to obtain a homogeneous group for observation, as well as ore objective final conclusions. All diagnoses from their previous history of any psychotic episodes (for example paranoidal or depressive syndromes) were criteria for exclusion.

The criteria for exclusion from the study in the group of SUD patients (without MANIA) were a medical history of psychosis (schizophrenic, affective), significant mood and/or anxiety disorders that required pharmacological treatment, and intellectual disability or genetic, severe, or uncompensated somatic (endocrinological, cardiovascular, renal, neoplastic, autoimmune, cachexia) or organic (with a manifestation of epilepsy) diseases. The healthy controls had normal intellectual skills, free of any psychoactive substances addictions (SPA), use of SPA in risky and harmful pattern (both present and past), as well as somatic and psychic disorders.

Table 1 shows a detailed, total and percentage, distribution of the examined types of addiction in all SUD patients with the SUD MANIA type distinguished. After the approval of the Bioethics Committee of the Pomeranian Medical University in Szczecin (KB-0012/106/16) had been granted, an informed, written consent agreement for the participation in the study was obtained. The investigation was performed in the Independent Laboratory of Health Promotion at the PUM University. The drug-addicted participants diagnosed with a maniacal episode (SUD MANIA) were recruited after a minimum of three months of controlled abstinence from drugs.

### 2.2. Measures

All participants in the study underwent a psychic and physical examination to exclude any somatic or organic disturbances. A structured psychiatric interview was carried out to assess the proper diagnosis of the patients according to DSM-IV (American Psychiatric Association APA DSM-IV criteria) and ICD-10 (ICD-10 classification of Mental and Behavioral Disorders: Clinical Description and Diagnostic Guidelines) criteria [84]. The patients with polysubstance use disorder with and without mania as well as healthy control individuals were clinically examined by a psychiatrist. The Mini International Neuropsychiatric Interview (M.I.N.I.), the NEO Five-Factor Personality Inventory (NEO-FFI), and the State-Trait Anxiety Inventory (STAI) questionnaires were used as a helpful and an evaluating tool. 

The STAI test helps to estimate the intensity of anxiety in two areas, including anxiety traits (A-Trait), which may be understood as a lasting tendency to experience worry, stress, and a kind of discomfort, and anxiety states (A-states), connected with fear and a readiness of the autonomic nervous system to exhibit temporary arousal in response to particular situations [85]. 

The NEO Five-Factor Inventory (Personality Inventory, NEO-FFI) is composed of six components, with five traits for each of them: the neuroticism axis (anxiety, depression, hostility, impulsiveness, self-consciousness, and vulnerability to stress), the extraversion axis (activity, assertiveness, excitement seeking, gregariousness, positive emotion, and warmth), an axis of openness to experience (actions, aesthetics, fantasy, feelings, ideas, Values), the agreeableness axis (altruism, compliance, modesty, straightforwardness, tendermindedness, and Trust,) and the conscientiousness axis (striving for achievement, competence, deliberation, dutifulness, order, and self-discipline) [86]. 

The scores for both inventories—the NEO-FFI and STAI—were described by stens. The conversion of severe scores into the stens schedule was carried out in accordance with the Polish norms for adults in which 1–2 stens indicated a very low score, 3–4 stens a low score, 5–6 an average score, 7–8 a high score, and at least 9–10 stens a very high score.

Only those patients who fulfilled the criteria for a maniacal episode (diagnosed SUD MANIA) had an analysis made concerning a dependence between personality traits and *DRD2* rs6276 gene polymorphisms in relation to non-addicted controls.

### 2.3. Genotyping

The genomic DNA was extracted from venous blood by means of standard procedures. Genotyping was performed using the real-time PCR method (LightCycler^®^ 480 II System; Roche Diagnostic, Basel, Switzerland).

## 3. Results

The frequency of distributions was found in all examined patients diagnosed with SUD MANIA and SUD and the control subjects in accordance with the HWE (Table 2).

No difference was found for *DRD2* rs6276 genotype frequencies among the studied sample between the patients with a diagnosis of SUD comorbid with mania (SUD MANIA), with SUD without mania, and control healthy subjects (Table 3).

Table 4 shows values that had been counted for all the NEO Five-Factor Inventory scores as well as the STAI scale state presented as variants of interactions for SUD and SUD MANIA and healthy controls. Scores are expressed as mean values and standard deviations.

We found that the differences in the frequency of genotype and alleles for the *DRD2* rs6276 gene between the patients with SUD MANIA and patients with SUD and healthy controls were of no statistically significance. 

When we compared the healthy controls and the SUD MANIA patients, for the latter, we observed significantly higher results on the STAI trait scale (M 7.70 vs. M 5.16, *p* < 0.0001), the STAI state scale (M 6.28 vs. M 4.68, *p* < 0.0001), the Scale of Neuroticism in NEO FFI (M 7.17 vs. M 4.67, *p* < 0.0001), and the Scale of Openness in NEO FFI (M 5.30 vs. M 4.53, *p* = 0.0054).

When we compared the healthy controls with the SUD MANIA patients, we found significantly lower results on the Scale of Agreeability in NEO FFI (M 4.15 vs. M 5.60, *p* < 0.0001) and on the Scale of Conscientiousness in NEO FFI (M 5.13 vs. M 6.07, *p* < 0.001).

The analysis of the healthy controls and the SUD patients showed significantly higher results for the latter on the STAI trait scale (M 6.93 vs. M 5.16, *p* < 0.0001), the STAI state scale (M 5.74 vs. M 4.68, *p* < 0.0001), the NEO FFI Scale of Neuroticism (M 6.60 vs. M 4.67, *p* < 0.0001) and the NEO FFI Scale of Openness (M 4.89 vs. M 4.53, *p* = 0.0250).

The comparison of the healthy subjects with the SUD ones resulted in data showing that the latter had significantly lower results on the Extraversion Scale of NEO FFI (M 5.67 vs. M 6.37, *p* = 0.0004) and the Scale of Agreeability of NEO FFI (M 4.36 vs. M 5.60, *p* < 0.0001), as well.

Regarding the SUD MANIA group and patients with the only SUD and without mania, for this latter group, significantly higher results were found on the STAI state scale (M 6.28 vs. M 5.74, *p* = 0.0051) and on the Neuroticism Scale of NEO FFI (M 7.17 vs. M 6.60, *p* = 0.0345).

Compared with the healthy participants, all SUD patients had significantly lower results on the Scale of Conscientiousness of NEO FFI (M 5.13 vs. M 5.78, *p* = 0.0263).

The results of the 2 × 3 factorial ANOVA analysis of the NEO Five-Factor Personality Inventory (NEO–FFI) and the State-Trait Anxiety Inventory (STAI) sten values are summed up and presented in Table 5.

As regards interactions, a significant result was found for all the groups (SUD MANIA vs. SUD vs. controls) on the NEO FFI Neuroticism Scale and the proportion of *DRD2* rs6276 (Figure 2, F_4,589_ = 2.659, *p* = 0.0320) was 1.7% of the variance (Table 5). The post hoc test results are demonstrated in Table 6.

## 4. Discussion

Substance use disorders (SUDs) and related disorders with many psychopathological symptoms are still persistent public health issues [87]. It has been estimated that there are more than 3.4 billion illegal drug users, and this results in over 12% of all deaths per year (World Health Organization). The use of such substances as alcohol, marijuana, opioids, cocaine, etc. often begins during adolescence, which is a critical time for the process of development at different levels. The heightened motivation to obtain arousal from rewards is severe, as indicated by an increase in risk-taking behaviors. On the other hand, it has been proven that reward sensitivity, avoiding unpleasantness, and looking for pleasure are closely dependent on dopamine neurotransmission, and several active dopamine receptors [88,89,90,91]. In the prefrontal cortex, dopamine action is involved in the efficiency of cognitive functions [92,93], as well as in thoughts and emotional and behavioral processes using the meso–cortico–limbic pathway [94]. Finally, chronic substance abuse may lead to exaggerated dopamine turnover and a clinical response expressed in different psychiatric disturbances [95]. These disturbances include paranoid or affective disorders, which most often require medical treatment for their severe life-threatening course [83]. In the first-episode psychotic patients (FEP), it was observed that those with cannabis-associated psychosis presented a greater degree of positive symptoms and dissociation and were significantly worse in terms of overall functioning [96]. According to clinical observation, the authors concluded that cannabis use not only can be a predictor of FEP manifestation but may also influence the disease duration and outcome [96,97]. In cannabis users, in comparison to the non-user group, a process of improvement and recovery after a psychotic episode was found to worsen over time [96]. 

In our study, we found that stimulants predominated (82.1%) in the subgroup of 89 SUD MANIA men, whereas cannabinols (72.6%), mixed substances (63.2%), and alcohol (58.9%) were mostly related to manic syndrome. These data for SUD MANIA patients were very close to those obtained in a total of the examined SUD men, particularly concerning the percentage of the stimulants’ contribution as a dominating drug in both. On the other hand, an insignificantly low role of cocaine in the risk of SUD MANIA has been demonstrated (8.4% vs. 11.8%) (Table 1).

There is evidence that the heavy use of cannabis, which, besides alcohol, are most popular group of substances abused too freely, may lead to the increased risk of psychosis (usually paranoidal or affective type) and suicide attempts. However, their harmful effects on depression and anxiety still seem not to be as convincing [98]. Additionally, stimulants such as methamphetamine are more typical and are attributed to paranoidal psychoses [99,100], whereas alcohol usually aggravates depression and may relate to a “double diagnosis” (alcohol addiction and bipolar disorder) [101]. It is probable that the selected SUD MANIA patients in the current study were also characterized by an additional specific vulnerability, such as neuroticism, personality traits, or epigenetic factors [102], which influenced their clinical manic view in cannabinoid, stimulant, and alcohol addiction.

The proteins that contribute to dopaminergic signaling are believed to strongly participate in drug addiction [103,104,105]. Until now, several investigations have concentrated on determining the linkage between the addiction process and specific genes connected with the dopaminergic system—for example, genes for dopamine transporter (*DAT*), dopamine D2 receptor (*DRD2*), catechol-O-methyltransferase (*COMT*), and dopamine-β-hydroxylase (*DBH*), but unfortunately there are not many well-established results [106]. Neurotransmission is also indirectly regulated by the DAT receptors, which are in the presynaptic dopaminergic neurons. The action of DAT is based on terminating the dopamine signaling in the synapse through its reuptake from the synaptic space. Some stimulant drugs such as cocaine have a high affinity to DAT receptors and bind it, and cocaine inhibits dopamine reuptake [107]. Consequently, synaptic levels of dopamine are elevated, and this is experienced as the effect of a “high” after cocaine intoxication. The dopamine D2R receptor encoded by the *DRD2* gene is also critical in dopaminergic signaling. Therefore, the genetic mechanisms that may modify dopamine receptor sensitivity and central representation are important in the regulation of the dopaminergic pathway [105,108].

In humans, the *DRD2* gene is polymorphic to a high degree—there are 204 synonymous SNPs, 294 missense SNPs, 190 noncoding transcript variant SNPs, and 22,935 intronic SNPs [109]. 

Although active coding SNPs have a noticeable influence on the encoded proteins, especially these variants situated in the noncoding genome regions, they are less predictable. It has been shown that over 80% of the SNPs in the human genome are recognized as responsible for some traits or diseases that are in the noncoding parts of the genome, including promoters, silencers, enhancers, and others [110]. The main role of these regulatory SNPs (rSNPs) is probably to modify the gene expression in a way that is specific to its allele. Therefore, they are thought to be a valuable class of genetic variation that may have an essential function in creating multiple traits. Recently, it has been verified that SNPs in the 3′UTR of genes are capable of affecting miRNA bindings, and as a result, the expression of genes for specific targets also for the modification of existing binding sites or the formation of new binding sites becomes possible [111,112]. This group of SNPs, which are identified as miRNA-binding SNPs or miRSNPs, have started to be classified in the human genome as a new separate group of rSNPs. They have an important role in the ethicology of some chronic, life-threatening diseases such as cancer [113], neurodegeneration [114], and hypertension [115]. What is known about miRNAs is that they are built of small noncoding RNAs (approximately 21 nucleotides) that are able to diminish the efficacy of the translation stage or stimulate a process of mRNA degradation through the base-pairing of their seed region (nucleotides 2–7 from the 5′ end of the mature miRNA) by means of complementary sequences that are typically located next to the 3′-untranslated region (UTR) of target genes [116]. Growing evidence shows that the miRNA in gene regulation seems to be crucial for mechanisms underlying the causes of numerous human diseases [117]. 

Crocco et al. (2015) showed that rs6276 of *DRD2* belongs to the group of miRSNPs. In their in silico analysis, they used six different miRNAs that were predicted to bind to the sequence containing polymorphism rs6276. The miR-485-5p was one of them. The authors stated that the highest energy of a binding level was to 3′UTR with the minor G allele. Therefore, G allele is probably associated with an increased miRNA–mRNA binding, and this mechanism may hypothetically lead to a stricter repression of *DRD2* expression [118]. It cannot be excluded that a mechanism of down-regulation of *DRD2* expression, which was described in striatal and extrastriatal brain regions in the elderly [118,119], might have had their source in regulations, as mentioned above.

SUD is a very complex phenotype of unexplained etiology. In our study, we found no difference in *DRD2* rs6276 genotype frequencies in the studied sample between patients diagnosed with SUD MANIA and those with SUD, as well as healthy SPA-free control subjects. These results demonstrate that the *DRD2* rs6276 genotype variant, documented in other data as of importance with regard to probable disturbances [120,121], seems to be of no special significance in addicted patients (SUD)—independently of their affective (according to a temporal study, “maniacal”) disturbances.

A possible explanation of this finding might be a complex etiology of substance use disorders (SUD), since it involves environmental factors that are linked to specific effects of psychoactive substances and genetic variations that influence and regulate each other, as explained by epigenetic hypotheses. The latter could explain why the risk for developing an addiction may be up to 40–60% [105]. However, these substances are yet to be regulated.

Genetic factors determine many human traits and diseases, though the range of genetic effects remains unknown [122]. By addressing the “mystery of missing heritability” Zuk et al. (2012) imply that relationships between loci (epistasis) might partially explain the gap that is present between a genetic risk factors that are determined and all heritability factors which that been estimated so far [123]. The interactions, both between gene and gene and between gene and environment, are predicted have an important role, but there is a scanty number of well-proven studies demonstrating interacting genes with a strong impact on complex traits [124,125,126,127] or their dependence on exogenous factors [128,129].

It is obvious that most people in the general population have the opportunity to try any of various psychoactive substances (for example alcohol) during their lives, but only a small part of them develop SUDs [130]. Strong evidence indicates that vulnerability to SUD is genetically determined. Epidemiological studies based on families and twins show that the heritability of SUD may be estimated at roughly 50% [131].

Although the contribution of genetic factors to SUD has been demonstrated, efforts to identify the specific SUD-associated genes with traditional molecular genetics methods were mostly unsuccessful. This can especially be observed in searching for genes designated as being crucial in neuropsychiatric disorders. The explanations for this are twofold: (i) there is a strong environmental influence on the immediate genetic effect, and (ii) the causal linkage of the gene and the disease is probably too long and/or complex for direct observation [132]. The introduction of the concept of “endophenotype” has provided an invaluable approach that allows identifying all the genes that could help identify individuals who suffer from or are threatened by psychiatric disturbances or, ultimately, mental diseases. The endophenotype concept is understood more as a simple clue to understanding the pathological syndrome referred to in the genetic data. Therefore, a genetic analysis should be conducted only in relation to clinical and cytological variables and areas such as those involved with cognitive functions, neurophysiological activity, anatomical structure, and biochemical regulations [133].

Our study shows that selected personality features can be used as endophenotype markers of SUD; thus, a trial to find a connection between genes and SUD, and consequently a better understanding of the possible peculiar differences, may have practical value for achieving a better understanding of the vulnerability and/or resilience to SUD.

In our study, differences were found in personality traits between subjects with SUD MANIA, with SUD without mania, and healthy controls. The SUD patients with a maniacal syndrome yielded significantly higher results on the Neuroticism Scale and higher levels of anxiety (STAI state results) compared to the patients with SUD without mania. Simultaneously, SUD MANIA patients yielded lower results on the Conscientiousness Scales compared to SUD patients (Table 4). SUD patients had significantly higher results on both the Neuroticism and Openness scales and higher levels of anxiety (STAI state and trait results) compared to healthy controls. However, SUD patients yielded lower results on the Extraversion and Agreeability scales compared to controls. These findings are consistent with the results commented on by other authors showing that people with substance use disorders may have a common personality profile, i.e., higher expression of neuroticism, lower self-conscientiousness, and lower abilities in agreeableness [134,135].

Neuroticism a personality trait that is characterized by a rigid tendency to excessively perceive typical and ordinary situations as being life-threatening and difficult to manage. Some neurotic characteristics are included in almost all other models of personality traits [136,137]. It is also a distinct trait that systematically varies within and between populations. Typically, the across-population variation for neuroticism stands at 3 to 10% of the total variation [138,139,140]. It is worth noting that the higher neuroticism scores observed in the actual study are congruent with other attainable data showing that neuroticism is related to some of the most common mental disorders, including mood, anxiety, and substance-use disorders [141,142,143].

In our study, the neuroticism scores were significantly statistically differentiated for all the subgroups (Table 4). SUD MANIA subjects had significantly higher scores on the Neuroticism Scale of NEO FFI compared to patients with SUD and controls (M 7.17 vs. M 6.60, *p* < 0.035 and M 7.17 vs. M 4.67, *p* < 0.000, respectively). The scores between SUD subjects and controls were also significantly significant (M 6.60 vs. M 4.67, *p* < 0.000 respectively) on the same scale. These data show that healthy controls had the lowest values of neuroticism. The data are consistent with clinical observations—namely that a maniac state is characterized by higher openness, but lower conscientiousness and agreeableness, which is very similar to SUD patients—although patients from both groups are free of SPA at the examination stage. We could formulate a hypothesis that neuroticism, analyzed as a kind of personality construct, may be one of the main conditions that predispose individuals to SUD syndrome or, at least, maniacal syndrome in the course of SUD. Furthermore, it may show that a more neurotic personality seems to be continuously independent of the psychotic state in manic episodes. This could help differentiate manic episodes resulting from the endogenous disease, developed due to chronic SPA intoxication. As is the case in endogenous psychoses, the personality is pathologically changed, and the way to determine how is not clear because psychotic symptoms dominate other deviations.

It may be supposed that patients with serious psychiatric comorbidities are more susceptible to the occurrence of psychical disease as psychosis or mania to stimulant-mediated factors in the course of probable underlying neurotransmitter dysregulation. However, in numerous cases of stimulant-induced psychoses reported to the United States Food and Drug Administration, no preexisting psychiatric disturbances or substance addiction history were recognized [17,23,24,144,145,146].

There is a body of evidence that explains that different variations in genes responsible for regulating the dopaminergic brain system modulate the processing of differentiation—which is centrally involved with decision-making, reinforcement learning, and risk assessment—and are located in the striatum and areas of the prefrontal cortex and are widespread in the limbic system [64]. Dopamine has been proved to have a role as being one of the main neurotransmitters in the system of behavioral approach [147,148]. Multiple correlational and experimental methods have shown the relevance of dopamine in extraversion and neuroticism in behavioral components [149,150]. Additionally, some variants in genes for dopamine are associated with variations in characteristic personality traits, although the conclusions of these associations seem to be less clear [151,152]. However, it is obvious fact that dopamine is involved in regulating brain systems that control cognitive and emotional decision processes underlying neuroticism [140,150]. This brief observation may support the opinion that not only the constitutional features but also the environmental conditions may have a strong influence on gene expression [153].

*DRD2* is probably a crucial gene that is associated with cognitive processes, mental disorders, and drug addiction [154,155,156,157,158], though the statistical evidence for its causative relationships is not persuasive enough [105]. The dopamine D2R receptor, which is encoded by the *DRD2* gene, is involved in the regulation of several brain functions responsible for perception, feeling, realization, control such as locomotion and movement, cognition functions, mood and affect, and motivation dependent on reward [159,160]. The D2R receptors are highly expressed in the basal ganglia, especially in the striatal caudate nucleus and in the nucleus accumbens. The study of *DRD2* expression shows that *DRD2*-regulated local circuits in pyramidal neurons and interneurons of the prefrontal cortex may participate in the regulation of emotional and cognitive functions.

Moreover, *DRD2*, which is expressed in a cortical area, is involved with the emotional, sensory, and motor modalities. Therefore, it is possible that both dopamine and DRD2 act as synchronizing signals for different activities connecting these regions in order to regulate behavioral modalities [161].

Sullivan et al. (2013) identified a promoter SNP of *DRD2* responsible for its enhanced expression (rs12364283) and two SNPs of intronic *DRD2* flanking exon 6, rs2283265 and rs1076560, which reduce the formation of D2S [105,158]. Research on healthy adults has shown that the rs6277 (C957T) polymorphism of the *DRD2* and the rs1800497 (Taq1A) polymorphism of the neighboring gene ankyrin repeat and kinase domain containing 1 (ANKK1) may alter dopaminergic signaling and influence prefrontal executive functions [32].

To date, there have been no data in the literature on the relationship between rs6276 in the *DRD2* gene with the neuroticism trait in patients with substance use disorder comorbid with mania. This is the first report regarding this issue. 

The main finding of our study is the significant interactions in the groups (SUD MANIA vs. SUD vs. Controls) between neuroticism and rs6276 SNP in the 3′ untranslated region in *DRD2*. The G/G homozygous variants were linked to lower results on the neuroticism scale in the SUD MANIA group (Figure 2 and Table 6). In our research study, G/G alleles had a protective effect on the intensity of neuroticism in the SUD MANIA group.

Other research studies showed that the combination of two highly prevalent *DRD2* SNPs, rs6276 and rs6277, decreased the expression of *DRD2* as a consequence of a diminished *DRD2* mRNA, and moreover, a decreased D2R receptor availability and affinity [70,72,162,163,164,165]. Han et al. found that these combined effects of *DRD2* and SNPs also diminished the levels of phosphorylated β-catenin, presumably causing more β-catenin to be successfully degraded, and in this way, the transcription of the Wnt/β-catenin pathway of the target genes was stimulated. The Wnt/β-catenin regulation is thought to be one of the most stable signaling pathways in different species and serves its basic role in the processes of proliferation, development, and unfortunately diseases [166]. Disruption in Wnt signaling has been described in numerous pathophysiological states such as different types of cancer, neurodegenerative diseases, and metabolic disorders [167]. Han et al. showed that *DRD2* is a candidate for being a transcriptional modulator in the signal transduction of Wnt/β-catenin, with a wide spectrum of positive implications for health with a perspective of new therapeutics [166].

As far as we know, mental health is still an important problem in public health, which concerns both cognitive and emotional levels. According to the World Health Organization, health in general is understood as complete physical, social, and psychological wellbeing. This issue has been studied extensively. However, studies in the field of mental and general health have proved that they are tied very closely with a relationship with personality traits and environmental psychological factors [168,169,170,171].

It is assumed that personality traits are mostly heritable [172], while genetic variations in the process of the production, turnover, and reuptake of neurotransmitters such as dopamine have regulatory and connective roles [149,173]. Through a variety of methods, dopamine has been experimentally examined in relation to differences in personality traits [150]. However, the relationships between genes responsible for regulating dopamine activity and phenotypes of personality have been not consistent [151]. These results may be caused by a large number of vectors connected with the environmental factors—for example, a kind of parental support, every negative life event, and resource availability—which also affect the development and modulation of personality traits, resulting in complex phenotypes regardless of close or similar genotypes and at least the dependence on environmental circumstances [140].

## 5. Conclusions

We found a significant result for the groups (SUD MANIA vs. SUD vs. controls) on the NEO FFI Neuroticism Scale and *DRD2* rs6276 (Figure 2, F_4,589_ = 2.659, *p* = 0.0320), which accounted for 1.7% of the variance. The G/G homozygous variants were linked to lower results on the neuroticism scale in the SUD MANIA group, and G/G alleles had a protective effect on the level of neuroticism in the SUD MANIA group. The combination of personality traits and genetic factors with harmful SPA intoxication may show that the addiction mechanism generally and its pathogenesis in particular will be better understood and treated. Unfortunately, an analysis of the rs6276 variant in the *DRD2* gene is unlikely important enough to cause addiction itself—as was our supposition. Therefore, genotyping of the *DRD2* gene in correlation with other numerous hereditary determinants is required to determine the most specific markers and support the addiction recovery process.

## Figures and Tables

**Figure 1 ijerph-19-09955-f001:**
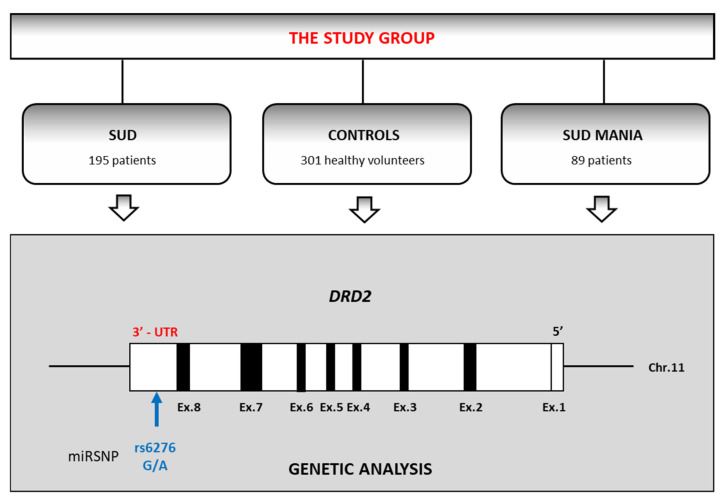
The flowchart of the study design (SUD—patients diagnosed with substance use disorder; SUD MANIA—patients with diagnosis substance use disorder comorbid with maniac episodes). *DRD2* gene is drawn to scale with black boxes representing exons and white boxes representing untranslated regions.

**Figure 2 ijerph-19-09955-f002:**
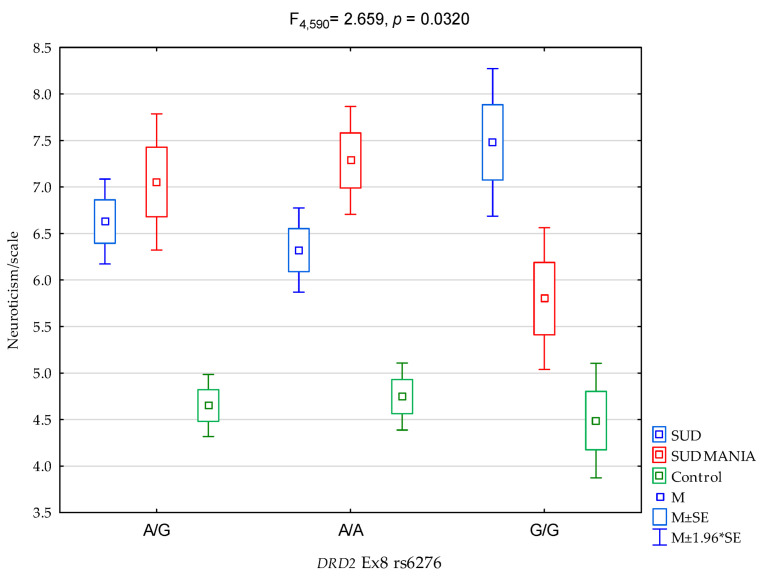
The relationship between patients diagnosed with polysubstance use disorder comorbid with a maniacal syndrome (SUD MANIA)/control and *DRD2* rs6276 and the NEO FFI Neuroticism scale.

**Table 1 ijerph-19-09955-t001:** The sorts of psychoactive substances used in all addicted subjects under study.

Sort of Substance/Type of Addiction	All Patients Diagnosed with SUD (*n* = 195)		All Patients Diagnosed with SUD MANIA (*n* = 89)	
	*n*	%	*n*	%
Behavioral addiction	80	41.0	45	45.3
Designer drugs	49	25.1	26	22.1
F10.2—alcohol	103	52.8	57	58.9
F11.2—opiates	37	19.0	16	22.1
F12.2—cannabinols	136	69.7	65	72.6
F13.2—sedatives and hypnotics	23	11.8	10	14.7
F14.2—cocaine	23	11.8	13	8.4
F15.2—stimulants	160	82.1	73	82.1
F16.2—hallucinogens	18	9.2	9	13.7
F19.2—mixed addictions	105	53.8	52	63.2

**Table 2 ijerph-19-09955-t002:** Hardy–Weinberg’s analysis for patients diagnosed with polysubstance use disorder comorbid with a maniacal syndrome (SUD MANIA) for patients diagnosed with substance use disorder without mania (SUD) and healthy control subjects.

Hardy–Weinberg Equilibrium Calculator Including Analysis for Ascertainment Bias		Observed (Expected)	Allele Freq	Test χ^2^	
				χ^2^	*p*
*DRD2 rs6276* SUD MANIA	A/A	42 (41.1)	A = 0.68 G = 0.32	0.181	>0.05
	A/G	37 (38.7)			
	G/G	10 (9.1)			
*DRD2 rs6276* SUD	A/A	85 (86)	A = 0.66 G = 0.34	0.103	>0.05
	A/G	89 (87)			
	G/G	21 (22)			
*DRD2 rs6276* control subjects	A/A	127 (121.8)	A = 0.64 G = 0.36	1.655	>0.05
	A/G	129 (139.3)			
	G/G	45 (39.8)			

*p*—statistical significance χ^2^ test.

**Table 3 ijerph-19-09955-t003:** Frequency of genotypes of the *DRD2* rs6276 gene polymorphisms estimated in SUD patients with mania SUD MANIA, patients with only SUD and without mania, and healthy controls.

Group	*DRD2 rs6276*				
	Genotypes			Alleles	
	A/A *n* (%)	A/G *n* (%)	G/G *n* (%)	A *n* (%)	G *n* (%)
A: SUD MANIA *n* = 89	42 (47.2)	37 (41.6)	10 (11.2)	121 (68.0)	57 (32.0)
B: SUD *n* = 195	85 (43.6)	89 (45.6)	21 (10.8)	259 (66.4)	131 (33.6)
C: Control *n* = 301	127 (42.2)	129 (42.9)	45 (15.0)	383 (63.6)	219 (36.45)
χ^2^ (*p* value)	A/B: 0.71 (0.701) A/C: 1.09 (0.578) B/C: 1.82 (0.403)			A/B: 0.14 (0.713) A/C: 1.14 (0.286) B/C: 0.81 (0.369)	

*n*—number of subjects.

**Table 4 ijerph-19-09955-t004:** The results of sten obtained in the STAI and NEO Five-Factor Inventory (NEO FFI) tests for healthy controls and all patients with a diagnosis of polysubstance use disorder and a maniacal syndrome (SUD MANIA) and patients with substance use disorder without mania (SUD).

STAI/NEO FFI	A: SUD MANIA (*n* = 89)	B: SUD (*n* = 195)	C: Control (*n* = 301)	A/C: Z (*p*-Value)	B/C: Z (*p*-Value)	A/B: Z (*p*-Value)
STAI trait/scale	7.70 ± 2.35	6.93 ± 2.24	5.16 ± 2.17	8.540 (0.0000 *)	7.887 (0.0000 *)	1.864 (0.0623)
STAI state/scale	6.28 ± 2.35	5.74 ± 2.42	4.68 ± 2.14	5.766 (0.0000 *)	4.871 (0.0000 *)	2.802 (0.0051 *)
Neuroticism/scale	7.17 ± 1.92	6.60 ± 2.23	4.67 ± 2.01	8.972 (0.0000 *)	8.844 (0.0000 *)	2.114 (0.0345 *)
Extraversion/scale	5.95 ± 2.34	5.67 ± 2.04	6.37 ± 1.97	−1.684 (0.0921)	−3.519 (0.0004 *)	0.703 (0.4816)
Openness/scale	5.30 ± 2.19	4.89 ± 1.94	4.53 ± 1.61	2.779 (0.0054 *)	2.240 (0.0250 *)	1.309 (0.1906)
Agreeability/scale	4.15 ± 2.05	4.36 ± 1.87	5.60 ± 2.09	−5.417 (0.0000 *)	−6.647 (0.0000 *)	−0.861 (0.3889)
Conscientiousness/scale	5.13 ± 2.22	5.78 ± 2.27	6.07 ± 2.15	−3.403 (0.0007 *)	−1.635 (0.1020)	−2.222 (0.0263 *)

*p*-value of statistical significance in Mann–Whitney U-test; *n,* number of subjects; M ± SD, mean ± standard deviation; * differences which are statistically significant (*p* < 0.005).

**Table 5 ijerph-19-09955-t005:** Differences in *DRD2* rs6276 and NEO Five-Factor Inventory, STAI scale between healthy control subjects and SUD and SUD MANIA.

STAI/NEO Five-Factor Inventory	Group	*DRD2* rs6276			ANOVA		
		A/A *n* = 251	A/G *n* = 260	G/G *n* = 80	F (*p*-Value)	η^2^	Power (Alfa = 0.05)
STAI trait/scale	SUD MANIA; *n* = 95	7.71 ± 2.21	7.64 ± 2.61	6.40 ± 1.07	F_4,589_ = 1.379 (*p* = 0.2397)	0.009	0.431
	B: SUD; *n* = 195	6.87 ± 2.28	6.80 ± 2.25	7.60 ± 2.04			
	C: Control; *n* = 301	5.26 ± 2.21	5.07 ± 2.03	5.16 ± 2.51			
STAI state/scale	SUD MANIA; *n*= 95	6.38 ± 2.41	6.03 ± 2.58	6.80 ± 0.63	F_4,589_ = 0.963 (*p* = 0.4274)	0.006	0.306
	B: SUD; *n* = 195	5.61 ± 2.48	5.70 ± 2.50	6.32 ± 1.97			
	C: Control; *n* = 301	4.97 ± 2.18	4.45 ± 2.00	4.58 ± 2.36			
Neuroticism/scale	SUD MANIA; *n*= 95	7.29 ± 1.91	7.05 ± 2.27	5.80 ± 1.23	F_4,589_ = 2.659 (*p* = 0.0320 *)	0.017	0.742
	B: SUD; *n* = 195	6.32 ± 2.15	6.628 ± 2.30	7.48 ± 2.02			
	C: Control; *n* = 301	4.75 ± 2.07	4.65 ± 1.93	4.49 ± 2.11			
Extraversion/scale	SUD MANIA; *n*= 95	5.83 ± 2.27	6.19 ± 2.62	5.60 ± 1.51	F_2,389_ = 1.434 (*p* = 0.2213)	0.010	0.447
	B: SUD; *n* = 195	6.02 ± 2.02	5.52 ± 1.95	5.08 ± 2.33			
	C: Control; *n* = 301	6.67 ± 1.99	6.05 ± 2.00	6.47 ± 1.75			
Openness/scale	SUD MANIA; *n*= 95	5.48 ± 2.09	5.27 ± 2.49	4.70 ± 1.34	F_2,389_ = 0.231 (*p* = 0.9208)	0.002	0.100
	B: SUD; *n* = 195	4.86 ± 1.84	4.99 ± 2.06	4.56 ± 1.85			
	C: Control; *n* = 301	4.51 ± 1.72	4.63 ± 1.46	4.31 ± 1.68			
Agreeability/scale	SUD MANIA; *n*= 95	4.42 ± 2.15	3.54 ± 1.83	5.20 ± 1.87	F_2,389_ = 1.221 (*p* = 0.3008)	0.008	0.384
	B: SUD; *n* = 195	4.32 ± 1.66	4.35 ± 1.98	4.56 ± 2.18			
	C: Control; *n* = 301	5.51 ± 2.22	5.50 ± 1.94	6.13 ± 2.06			
Conscientiousness/scale	SUD MANIA; *n*= 95	4.90 ± 2.22	5.38 ± 2.24	5.20 ± 2.35	F_2,389_ = 0.558 (*p* = 0.6933)	0.004	0.186
	B: SUD; *n* = 195	5.66 ± 2.37	5.92 ± 2.32	5.68 ± 1.75			
	C: Control; *n* = 301	6.19 ± 2.27	5.96 ± 2.10	6.11 ± 1.92			

*—significant result.

**Table 6 ijerph-19-09955-t006:** Post hoc analysis of dependences between patients diagnosed with polysubstance use disorder comorbid with a maniacal syndrome (SUD MANIA)/control and *DRD2* rs6276 and NEO FFI Neuroticism scale.

*DRD2* rs6276 and NEO FFI Neuroticism Scale									
	{1} M = 5.52	{2} M = 6.02	{3} M = 5.08	{4} M = 6.19	{5} M = 5.83	{6} M = 5.60	{7} M = 6.05	{8} M = 6.67	{9} M = 6.47
SUD *DRD2* rs6276 A/G {1}		0.0934	0.3430	0.0888	0.4006	0.9010	0.0539	0.0000 *	0.0102 *
SUD *DRD2* rs6276 A/A {2}			0.0427 *	0.6791	0.6219	0.5360	0.9340	0.0236 *	0.2380
SUD *DRD2* rs6276 G/G {3}				0.0367 *	0.1454	0.4972	0.0310 *	0.0004 *	0.0068 *
SUD MANIA *DRD2* rs6276 A/G {4}					0.4407	0.4193	0.7085	0.2095	0.54130
SUD MANIA *DRD2* rs6276 A/A {5}						0.7459	0.5577	0.0220 *	0.1495
SUD MANIA *DRD2* rs6276 G/G {6}							0.5063	0.1120	0.2261
control *DRD2* rs6276 A/G {7}								0.0152 *	0.2360
control *DRD2* rs6276 A/A {8}									0.5682
control *DRD2* rs6276 G/G {9}									

*—significant statistical differences, M—mean.

## Data Availability

Not applicable.

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
