# Peer review of "Association of Polymorphism within the Putative miRNA Target Site in the 3′UTR Region of the DRD2 Gene with Neuroticism in Patients with Substance Use Disorder"

_ijerph, 2022, doi:10.3390/ijerph19169955_

Round 1
Reviewer 1 Report
I found this a very interesting, break-through study.
Nevertheless, I would have some suggestions about possible adjustments. For example, the article seemed a little too long, with many lengthy sentences and a lot of information not directly related to the aim of the study, making it hard to follow. A shorter, more goal-oriented paper would be easier to understand and catchier for the readers.
Moreover, an English check would be necessary to correct some minor spelling and sentences constructs mistakes.
The statement about the relation between cannabis use and psychosis at line 314-315 would need some reference (for examples may see: https://pubmed.ncbi.nlm.nih.gov/34688166/ or https://pubmed.ncbi.nlm.nih.gov/34886357/).
Finally, the methods were very well presented, with well-built study design and statistical analysis.
Author Response
Thank You for all Your suggestions.
Moreover, an English check would be necessary to correct some minor spelling and sentences constructs mistakes.
After Your remarks, we analyzed our test once again and tried to make it more clear and easier to understand and catchier for the readers. Most of the longest and to complicated sentences were verified and improved that way as it was possible.
The statement about the relation between cannabis use and psychosis at line 314-315 would need some reference (for examples may see: https://pubmed.ncbi.nlm.nih.gov/34688166/ or https://pubmed.ncbi.nlm.nih.gov/34886357/).
According to You suggestion we fulfilled the statement about the relation between cannabis use and psychosis at line 314-315 (in earlier version) /325-331 and 339-341 (in the latest one). Now, it really seems to be more complete as their evident differences in the intercourse of psychosis and global functioning in patients with SPA (cannabis) abuse and without them [References 171, 172].
Reviewer 2 Report
The study from Boroń and colleagues aimed to investigate associations between polymorphism rs6276 in the putative miRNA target site in the 3'UTR region of the DRD2 gene in patients with substance use disorder comorbid with a mania episode and selected personality features.
The topic is stimulating, and the manuscript provides results that could expand our knowledge of the relations between SUD, mania, and genetic regulation. However, I think it suffers from some limitations that authors must work out. Here are some points.
1) As a general comment, the introduction and, especially, the discussion is overly lengthy and somewhat unfocused. Authors should rethink the discussion, make it easier to follow, and eliminate some redundant parts. For example, the authors present the main finding at 3/4 of the entire section (line 485), while the reader expects it at the beginning.
2) Authors hypothesize that neuroticism analyzed as a kind of personality construct may be one of the leading conditions predisposing to SUD syndrome or, at least, maniac syndrome during SUD. Then, they assert that the main finding of their study is a significant interaction for the groups (SUD MANIA vs. SUD vs. Controls) between neuroticism and rs6276 SNP in the 3' untranslated region of DRD2. In particular, they observed that the G/G homozygous variants were linked to lower neuroticism scores in the SUD MANIA group. On this basis, the authors state that G/G alleles had a protective effect on the level of neuroticism in the SUD MANIA group. But, this statement contradicts the hypothesis of neuroticism as a predisposing factor for maniac syndrome during SUD. The authors must clarify this point.
3) de Marco et al. have recently published a systematic review of genetic polymorphisms associated with bipolar disorder comorbid to substance abuse (PMID: 35893041) that seems highly relevant for the introduction of the present manuscript.
4) Line 389, please substitute "determined" with a less deterministic term, such as, i.e., "regulated".
5) Lines 442-443, what do you mean with deviations? Please change it to make it more straightforward.
Author Response
Thank You for all Your suggestions.
1) As a general comment, the introduction and, especially, the discussion is overly lengthy and somewhat unfocused. Authors should rethink the discussion, make it easier to follow, and eliminate some redundant parts. For example, the authors present the main finding at 3/4 of the entire section (line 485), while the reader expects it at the beginning.
Thank you for your remarks which suggested us to check and correct our article once again and to the introduction and the discussion a little more focused and easier to follow. In our opinion it is now too complicated to make the test shorter as it might be not so understandable. We are anxious of fact that the subject of presented material is complex in relation to genetic and psychic regulations. This a reason why we presented the main finding at the entire section not at the very beginning. Although, we tried to do our best to make it so clear and corresponding with facts of the case as it is possible.
2) Authors hypothesize that neuroticism analyzed as a kind of personality construct may be one of the leading conditions predisposing to SUD syndrome or, at least, maniac syndrome during SUD. Then, they assert that the main finding of their study is a significant interaction for the groups (SUD MANIA vs. SUD vs. Controls) between neuroticism and rs6276 SNP in the 3' untranslated region of DRD2. In particular, they observed that the G/G homozygous variants were linked to lower neuroticism scores in the SUD MANIA group. On this basis, authors state that G/G alleles had a protective effect on the level of neuroticism in the SUD MANIA group. But, this statement contradicts the hypothesis of neuroticism as a predisposing factor for maniac syndrome during SUD. The authors must clarify this point.
Thank you for this comment. Yes indeed, in our study we obtained a result regarding one of the genotypes in the G/G homozygous variant as being associated with lower neuroticism scores in the SUD MANIA group. Based on this, the authors conclude that the G/G allele had a protective effect on neuroticism levels in the SUD MANIA group. We left the hypothesis of neuroticism as a predisposing factor for manic syndrome during SUD. Our reasoning stems from a far-reaching caution in our inference. What we see in G/G heterozygosity is extremely interesting, but it is a single genotype–out of many predisposing genes–as well as a relatively small group. Our dream is to increase the group–the project is underway. However, we are afraid to infer in a 0-1 fashion, as we are aware of multi-druginess and multi-factoriality. However, we see the validity of our analysis direction, and we thank you very much for this comment.
3) de Marco et al. have recently published a systematic review of genetic polymorphisms associated with bipolar disorder comorbid to substance abuse (PMID: 35893041) that seems highly relevant for the introduction of the present manuscript
We are grateful for Your suggestion which is expected to make our study more completed.
According to Your prompt we fulfilled the statement about the latest data concerning the relationships between genetic polymorphisms and bipolar disorder comorbid to substance abuse. We put some important information out of it at line 151-158 [Reference 173]
- Line 389, please substitute "determined" with a less deterministic term, such as, i.e., "regulated".
Thank you for this note as it was obvious language mistake in this part of dissertation. We of course substituted term "determined" with "regulated".
The correct sentence sounds now: “However, it is yet to be regulated” (line 423-424 of in the latest version).
5) Lines 442-443, what do you mean with deviations? Please change it to make it more straightforward.
Thank you for your doubts about the ideas which were expressed by authors in discussion in a way not enough clear.
Past: SUD MANIA subjects had significantly highest scores on the NEO Five Factor Inventory Scale of Neuroticism compared to patients with SUD (M 7.17 vs. M 6.60, p<0.035) and controls (M 7.17 vs. M 4.67, p<0.000). However, SUD subjects had significantly higher scores on the NEO Five Factor Inventory Scale of Neuroticism compared to controls (M 6.60 vs. M 4.67, p< 0.000).
We analyzed once again the results of our investigation and change this part of dissertation into the sentence:
Now (line 465 -469): SUD MANIA subjects had significantly highest scores on the Neuroticism Scale of NEO FFI compared to patients with SUD and controls (M 7.17 vs. M 6.60, p<0.035 and M 7.17 vs. M 4.67, p<0.000, properly). The scores between SUD subjects and controls also were significantly important (M 6.60 vs. M 4.67, p< 0.000 properly) in the same Scale. This data show that healthy controls had the lowest values of neuroticism.
